# Evaluation of African American Language Bias in Natural Language Generation

**Nicholas Deas**
Columbia University
Department of
Computer Science
ndeas@cs.columbia.edu

**Jessi Grieser**
University of Michigan
Department of Linguistics
jgrieser@umich.edu

**Shana Kleiner**
University of Pennsylvania
School of Social Policy and
Practice, Annenberg School for
Communications
skleiner@upenn.edu

**Desmond Patton**
University of Pennsylvania
School of Social Policy and
Practice, Annenberg School for
Communications
dupatton@upenn.edu

**Elsbeth Turcan**
Columbia University
Department of
Computer Science
eturcan@cs.columbia.edu

**Kathleen McKeown**
Columbia University
Department of
Computer Science
kathy@cs.columbia.edu

## Abstract

*Warning: This paper contains content and language that may be considered offensive to some readers.*

While biases disadvantaging African American Language (AAL) have been uncovered in models for tasks such as speech recognition and toxicity detection, there has been little investigation of these biases for language generation models like ChatGPT. We evaluate how well LLMs understand AAL in comparison to White Mainstream English (WME), the encouraged "standard" form of English taught in American classrooms. We measure large language model performance on two tasks: a counterpart generation task, where a model generates AAL given WME and vice versa, as well as a masked span prediction (MSP) task, where models predict a phrase hidden from their input. Using a novel dataset of AAL texts from a variety of regions and contexts, we present evidence of dialectal bias for six pre-trained LLMs through performance gaps on these tasks.

## 1 Introduction

Task-specific models proposed for speech recognition, toxicity detection, and language identification have previously been documented to present biases for certain language varieties, particularly for African American Language (AAL) (Sap et al., 2022; Koenecke et al., 2020; Meyer et al., 2020; Blodgett and O'Connor, 2017). There has been little investigation, however, of the possible language variety biases in Large Language Models (LLMs) (Dong et al., 2019; Brown et al., 2020; Raffel et al., 2020), which have unified multiple tasks through language generation.

While there are largely beneficial and socially relevant applications of LLMs, such as in alleviating barriers to mental health counseling[1] and medical healthcare (Hsu and Yu, 2022) access, there is also potential for biased models to exacerbate existing societal inequalities (Kordzadeh and Ghasemaghaei, 2022; Chang et al., 2019; Bender et al., 2021). Past algorithms used in psychiatry and medicine have been shown to be racially biased, in some cases leading to, for example, underestimating patient risk and denial of care (Obermeyer et al., 2019; Straw and Callison-Burch, 2020). Furthermore, LLMs capable of understanding AAL and other language varieties also raise important ethical implications, such as enabling increased police surveillance of minority groups (see Patton et al. 2020 and section 8 for further discussion). Therefore, it is necessary to investigate the potential language variety biases of language generation models to both increase accessibility of applications with high social impact and also anticipate possible harms when deployed.

Moreover, prior work (Grieser, 2022) has shown that African American speakers talking about race-related issues use language in ways which may draw on morphosyntactic features of AAL in order to subtly foreground the race aspect of the discussion topic without explicit mention. Most training corpora include little representation of AAL (see further discussion in section 3), and even those that do can still fail to capture its significant regional and contextual variation (see Farrington et al. 2021 for examples). Without the ability to interpret these

---

[1] https://www.x2ai.com/

subtler meanings of AAL, LLMs will undoubtedly exacerbate the misunderstandings which already take place between AAL speakers and other communities.

| | AAL | WME |
|---|---|---|
| **Source Text** | Since RED gone, my HEAD gone & dats thee ONLY shit WRK. | Since Red is gone, my head is gone, and that's the only thing working. |
| | **Model-Generated AAL** | **Model-Generated WME** |
| **ChatGPT Counterpart** | Since Red ain't around, my head ain't right, and that's the only thing keepin' me going. | Since Red left, my head is gone and that's the only thing that works. |
| **GPT-4 Counterpart** | Since Red gone, my head gone, and that's the only thing workin'. | Since Red left, my head hasn't been right and that's the only thing that works. |

Table 1: Examples of ChatGPT and GPT-4 counterpart predictions. Given text in either WME or AAL, models attempt a semantically-equivalent rewriting in the other language variety.

Given the lack of African American representation in LLMs and the possible harms to the AAL-speaking community, we focus on LLMs' understanding of AAL to investigate biases. AAL is a language variety which follows consistent morphological, syntactic, and lexical patterns distinct from WME, such as the dropped copula (e.g., "*she at work*") and aspect markers (e.g., the habitual *be*: "*he be running*") (Lanehart, 2001; Green, 2009). We use Grieser (2022)'s definition of AAL as the grammatically patterned variety of English used by many, but not all and not exclusively, African Americans in the United States. Following Baker-Bell (2020) and Alim and Smitherman (2012), we also use the definition of White Mainstream English (WME) as the dialect of English reflecting the linguistic norms of white Americans. While previous linguistic literature occasionally uses the terms "Standard American English" and "African American Vernacular English," we employ AAL and WME instead to avoid the implication that AAL and other language varieties are "non-standard" and to more precisely identify the demographics of prototypical WME speakers, similarly to Baker-Bell (2020) and Alim and Smitherman (2012). Examples of AAL and WME are shown in Table 1.

We evaluate understanding of AAL by LLMs through production of language in each variety using automatic metrics and human judgments for two tasks: a counterpart generation task akin to dialect translation (Wan et al., 2020; Harrat et al., 2019) (see examples in Table 1) and a masked span prediction (MSP) task where models predict a phrase that was removed from their input, similar to Groenwold et al. (2020). We summarize our contributions as follows: **(1)** we evaluate six pre-trained, large language models on two language generation tasks: counterpart generation between language varieties and masked span prediction; **(2)** we use a novel dataset of AAL text from multiple contexts (social media, hip-hop lyrics, focus groups, and linguistic interviews) with human-annotated counterparts in WME; and **(3)** we document performance gaps showing that LLMs have more difficulty both interpreting and producing AAL compared to WME; our error analysis reveals patterns of AAL features that models have difficulty interpreting in addition to those that they can understand.

## 2 Background: Bias

In measuring AAL understanding, we identify evidence of bias through performance gaps and analysis of model behavior with each language variety. Following Blodgett et al. (2020), findings of bias could result in both allocational harms and representational harms posed by the evaluated models[2].

While LLMs are becoming more available and valuable resources, the models' lack of understanding of AAL limits their use by AAL speakers, and this disparity will only grow as the use of these models increases across social spheres. Our evaluation attempts to quantify these *error disparities* (Shah et al., 2020) by measuring models' understanding of AAL and WME texts. When LLMs do not perform equally well on different language varieties, the LLM itself as a resource becomes unfairly allocated, and speakers of minoritized language varieties like AAL are less able to leverage the benefits of LLMs. AAL speakers would be particularly unfairly impacted with applications in areas of health, including mental health.

Additionally, our evaluation includes a qualitative analysis of how AAL is currently understood and produced by LLMs. Prior sociolinguistic works discuss and study how attitudes toward African American speakers have formed linguistic prejudices against AAL (Baker-Bell, 2020; Baugh, 2015), as well as how stereotyped uses of AAL by non-AAL speakers can perpetuate racial divides (Ronkin and Karn, 1999). Stereotypical or offensive uses of AAL by LLMs thus reflect a representational harm to AAL speakers that can further

---

[2]*Allocational harms* are reflected in the unfair distribution of resources and opportunities among social groups, while *representational harms* are reflected in disparate or harmful representations of a particular group (see Blodgett et al. (2020) for further discussion).

promote these views. We advocate for approaches which carefully consider sociolinguistic variation in order to avoid generation of inappropriate speech across different settings.

## 3 Data

Biases in pre-trained language models can often be attributed to common training datasets. Training corpora for LLMs are typically drawn from internet sources, such as large book corpora, Wikipedia, outbound links from Reddit, and a filtered version of Common Crawl[3] in the case of GPT-3 (Brown et al., 2020), which can severely under-represent the language of African Americans (Pew Research Center, 2018; Dolcini et al., 2021). Though few estimates of the presence of AAL in datasets exist, one study estimates that in the Colossal Cleaned Crawl Corpus (C4), only 0.07% of documents reflect AAL (Dodge et al., 2021). Beyond C4, African Americans are significantly underrepresented in data sources such as Wikipedia [4] (0.05%) and news articles (6%; Pew Research Center 2023), falling well below the national average. Additionally, as models learn from gold standard outputs provided by annotators, they learn to reflect the culture and values of the annotators as well.

### 3.1 Data Sources

There is significant variation in the use of features of AAL depending on, for example, the region or context of the speech or text (Washington et al., 1998; Hinton and Pollock, 2000). Accordingly, we collect a novel dataset of AAL from six different contexts. We draw texts from two existing datasets, the TwitterAAE corpus (Blodgett et al., 2016) and transcripts from the Corpus of Regional African American Language (CORAAL; Kendall and Farrington 2021), as well as four datasets collected specifically for this work: we collect all available posts and comments from r/BlackPeopleTwitter[5] belonging to "Country Club Threads", which designates threads where only Black Redditors and other users of color may contribute [6]. Given the influence of AAL on hip-hop music, we collect hip-hop lyrics from 27 songs, 3 from each of 9

---

[3]http://commoncrawl.org/
[4]https://meta.wikimedia.org/wiki/Community_Insights/Community_Insights_2021_Report/Thriving_Movement
[5]https://reddit.com/r/BlackPeopleTwitter
[6]To be verified as a person of color and allowed to contribute to Country Club Threads, users send in pictures of their forearm to reveal their skin tone.

Black artists from Morgan (2001) and Billboard's 2022 Top Hip-Hop Artists. Finally, we use the transcripts of 10 focus groups concerning grief and loss in the Harlem African American community and conducted as part of ongoing work by the authors to better understand the impacts of police brutality and other events on the grief experiences of African Americans. Following Bender and Friedman (2018), a data statement with further details is included in Appendix A.

50 texts are sampled from each dataset, resulting in 300 candidate texts in total. We use a set of surface level and grammatical patterns to approximately weight each sample by the density of AAL-like language within the text (patterns are listed in Appendix B). 12 additional texts are also sampled from each dataset for fine-tuning.

### 3.2 Data Annotations

Our interdisciplinary team includes computer scientists, linguists, and social work scientists and thus, we could recruit knowledgeable annotators to construct semantically-equivalent re-writings of AAL texts into WME, referred to as *counterparts*. The four human annotators included 2 linguistics students, 1 computer science student, and 1 social work scientist, all of whom self-identify as AAL speakers and thus have knowledge of the linguistic and societal context of AAL and racial biases. These annotators were familiar with both AAL and WME, allowing them to provide accurate annotations and judgements of model generations in both language varieties. Annotators were asked to rewrite the AAL text in WME, ensuring that the counterparts conserve the original meaning and tone as closely as possible (see Appendix C.1).

To compute inter-annotator agreement, we asked each annotator to label the 72 additional texts, and they also shared a distinct 10% of the remainder of the dataset with each other annotator. We compute agreement using Krippendorff's alpha with Levenshtein distance (Braylan et al. 2022; see Appendix D for more details) showing 80% agreement ($\alpha = .8000$). After removing pairs from the dataset where annotators determined that no counterpart exists, the final dataset consists of 346 AAL-WME text pairs including the 72 additional texts. Dataset statistics are included in Table 2.

| Dataset | # Samples | Avg. Length (AAL) | Avg. Length (WME) | Rouge-1 | Avg. Tox (AAL) | Avg. Tox (WME) |
|---|---|---|---|---|---|---|
| r/BPT Comments | 60 | 24.20 | 24.33 | 85.3 | 0.40 | 0.29 |
| r/BPT Posts | 61 | 8.67 | 9.95 | 81.2 | 0.08 | 0.08 |
| TwitterAAE (Blodgett et al., 2016) | 58 | 13.60 | 15.02 | 65.9 | 0.78 | 0.54 |
| CORAAL (Kendall and Farrington, 2021) | 56 | 13.07 | 13.34 | 84.1 | 0.16 | 0.09 |
| Focus Groups | 54 | 29.20 | 26.96 | 71.4 | 0.09 | 0.07 |
| Hip-Hop Lyrics | 57 | 9.39 | 10.58 | 67.8 | 0.47 | 0.40 |
| **AAL Total** | **346** | **16.23** | **16.60** | **77.4** | **0.33** | **0.25** |

Table 2: Characterization of the novel AAL dataset by text source including the number of text samples, length (in words) of aligned AAL and WME texts, Rouge-1 between aligned texts, and the average toxicity scores among dialects.

---

> *Translate the following African American Vernacular English into Standard American English:*
>
> *and ain't sixteen years old, this shit has got to stop. => And he is not sixteen years old, this shit has got to stop.*

Figure 1: Example prompt provided to GPT models in the counterpart generation task including the instruction and AAL text for which the model generates a WME counterpart (blue).

## 4 Methods

We evaluate multiple language generation models using two tasks. In the counterpart generation task, we evaluate models on producing near semantically-equivalent WME text given AAL text and vice versa to target LLMs' ability to interpret and understand AAL. A second task, masked span prediction, requires models to predict tokens to replace words and phrases hidden or *masked* from the input. This task resembles that of Groenwold et al. (2020), but spans vary in length and position. Much like BART pre-training (Lewis et al., 2020), span lengths are drawn from a Poisson distribution ($\lambda = 2$) and span locations are sampled uniformly across words in the original text. We independently mask noun phrases, verb phrases, and random spans from the text for more fine-grained analysis.

While our focus is on measuring model capabilities in *interpreting* AAL[7], these generation tasks allow us to test whether the model understands the language well enough to produce it. It is not our goal to produce a LLM that can generate AAL within the context of downstream tasks (see section 8 for further discussion).

### 4.1 Models

We consider six different models for the two tasks where applicable: **GPT-3** (Brown et al., 2020);

its chat-oriented successor, **ChatGPT (GPT-3.5)**[8]; **GPT-4** (OpenAI, 2023), currently OpenAI's most advanced language model ; **T5** (Raffel et al., 2020); its instruction-tuned variant, **Flan-T5** (Chung et al., 2022); and **BART** (Lewis et al., 2020). Flan-T5, GPT-3, ChatGPT, and GPT-4 are evaluated on the counterpart generation task, while GPT-3, BART, and T5 are evaluated on the MSP task. We note that the GPT models besides GPT-3 were not included in the MSP task because token probabilities are not provided by the OpenAI API for chat-based models. An example of the instruction provided to GPT models is provided in Figure 1. Notably, the instruction text provided to GPT models uses "African American Vernacular English" and "Standard American English" because prompts with these terms were assigned lower perplexity than "African American Language" and "White Mainstream English" by all GPT models, and lower perplexity prompts have been shown to improve task performance (Gonen et al., 2022). Additionally, GPT models are simply asked to translate with no additional instructions in order to examine their natural tendency for tasks involving AAL text. We evaluate both Flan-T5 fine-tuned on the 72 additional texts, referred to as *Flan-T5 (FT)* in the results, and Flan-T5 without fine-tuning (with automatic metrics only). Additional modeling details and generation hyperparameters are included in Appendices E.1 and E.2.

### 4.2 Metrics

We use both automatic and human evaluation metrics for the counterpart generation task. As with most generation tasks, we first measure n-gram overlap of the model generations and gold standard reference texts and in our experiments, we utilize the Rouge metric. In addition, to account for the weaknesses of word-overlap measures, we also measure coverage of gold standard references with BERTScore (Zhang* et al., 2020) using the

---

[7]In reference to model capabilities, "interpretation" and "understanding" refer to the ability of models to accurately encode the meaning and features of text in AAL and WME as opposed to cognitive notions of these terms.

[8]https://openai.com/blog/chatgpt/

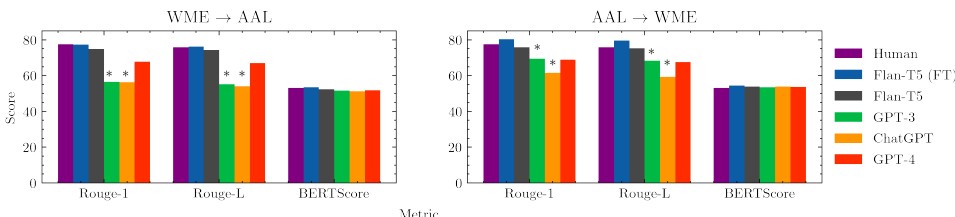

Figure 2: Automatic coverage metrics of model-generated AAL and WME counterparts. "Human" (purple) scores represent coverage metrics between the original AAL text and human-annotated WME counterparts. Significant differences between scores in the WME → AAL direction and in the AAL → WME direction are denoted by * (p≤ .05).

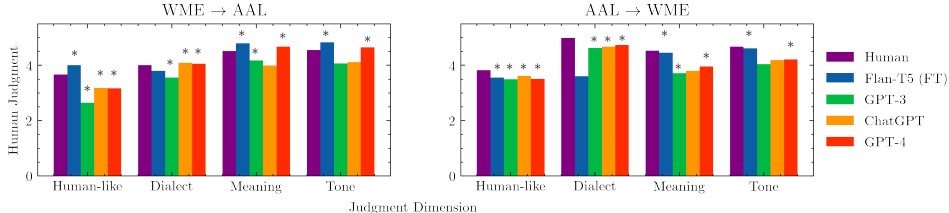

Figure 3: Human judgments for model-generated AAL and WME counterparts. "Human" (purple) scores represent judgments of original AAL text and human-annotated WME counterparts. Significant differences between scores in the WME → AAL direction and in the AAL → WME direction are denoted by * (p≤ .05). Flan-T5 without fine-tuning was evaluated with automatic metrics after human judgements were collected.

*microsoft/deberta-large-mnli* checkpoint, because it is better correlated with human scores than other models[9]. Specifically, original AAL is the gold standard for model-generated AAL and human annotated WME counterparts are the gold standard for model-generated WME. In some experiments, Rouge-1, Rouge-L, and BERTScore are presented as gaps, where scores for generating WME are subtracted from those for generating AAL. Due to the tendency of models to avoid toxic outputs and neutralize text, we also consider the percentage of toxic terms removed when transitioning from model inputs in one language variety to outputs in the other. Toxicity scores are derived as the number of words categorized as offensive in the word list of Zhou et al. (2021), and percent change between inputs and outputs are calculated as $\frac{(Tox_{in} - Tox_{out})}{Tox_{in}}$.

Human evaluation is also conducted on the generated counterparts. The same linguistics student, computer science student, and social work scientists involved in creating the dataset of aligned counterparts were also asked to judge model generations. As a baseline, human-generated counterparts are included in the human evaluation. 100 WME and AAL texts along with their generated

and annotated counterparts are randomly sampled from the dataset for human evaluation. We ensure that annotators do not rate human or model-generated counterparts for which they initially generated the WME counterpart. All annotators are asked to rate each assigned counterpart using 5-point Likert scales on the dimensions below.

*Human-likeness (Human-like)* measures whether the annotator believes that the text was generated by a human or language model. *Linguistic Match (Dialect)* measures how well the language of the counterpart is consistent with the intended English variety (i.e., AAL or WME). *Meaning Preservation (Meaning)* measures how accurately the counterpart conveys the *meaning* of the original text. And finally, *Tone Preservation (Tone)* measures how accurately the counterpart conveys the *tone* or other aspects beyond meaning of the original text. Additional details on the judgment instructions are included in Appendix C.2.

In the masked span prediction task, span predictions are evaluated using automated metrics: model perplexity of the reference span, and the entropy of the model's top 5 most probable spans. With the exception of GPT-3, experiments are repeated 5 times, randomly sampling spans to mask in each trial. Metrics are reported as the percent change in perplexity between WME and AAL.

[9] https://docs.google.com/spreadsheets/d/1RKOVpselB98Nnh_EOC4A2BYn8_2O1tmPODpNWu4w7xI/edit

# 5 Results

## 5.1 Are AAL and WME Metrics Comparable?

Studies such as Bugliarello et al. (2020) might suggest that in translation-like tasks, it is invalid to compare results from automatic metrics, such as BLEU, cross-lingually because: (1) different languages may use different numbers of words to convey the same meaning, and (2) models for different languages utilize different tokenization schemes.

Though we emphasize that AAL and WME are language varieties of English rather than distinct languages, a similar argument may be made that their Rouge scores are not directly comparable. However, the counterpart generation task setting does not suffer from either of the aforementioned weaknesses. To show this, we calculate differences in the number of words and 1-gram Type-Token Ratio for AAL and WME text pairs in our dataset.

As shown in Table 3, the total number of words in the AAL and WME texts are similar, and we find that the lengths of each pair of texts differ by less than 1/10th of a word (0.095) on average. Bugliarello et al. (2020) also finds that among metrics studied, translation difficulty is most correlated with the Type-Token Ratio (TTR) of the target language. Table 3 shows that the difference in the 1-gram TTR between AAL and WME is not statistically significant. Finally, as the same models are applied to both AAL and WME texts, the tokenization schemes are also identical. Therefore, the identified weaknesses of cross-lingual comparison do not apply to our results.

| Dialect | # Words | TTR |
|---------|---------|-----|
| AAL | 5632 | 0.274 (0.259, 0.290) |
| WME | 5665 | 0.260 (0.245, 0.275) |

Table 3: Comparison of Type-Token Ratios between the AAL and WME texts in the dataset. 95% confidence intervals calculated using the Wilson Score Interval are shown in parenthesis.

## 5.2 Counterpart Generation

Figure 2 shows results using automatic coverage metrics on counterpart generations in AAL and WME. Rouge-1, Rouge-L and BERTScore (the coverage scores) for model output are computed over the generated AAL or WME in comparison to the corresponding gold standards. We note that the

models consistently perform better when generating WME, indicating that it is harder for models to reproduce similar content and wording as the gold standard when generating AAL. ChatGPT is the worst model for producing WME from AAL, and ChatGPT and GPT-3 are nearly equally bad at producing AAL from WME. Flan-T5 (FT) does best for both language varieties, likely due to the fact that Flan-T5 (FT) was directly fine-tuned for the task. Flan-T5 without fine-tuning performs comparatively with slightly lower coverage scores in both directions. We also compute the coverage scores between the original AAL text from the dataset and the human annotated counterparts in WME, labeled as "Human" in Figure 2. Models tend to generate counterparts with lower coverage scores than the text input to the model, which reflects the alternative language variety. This suggests that it is difficult for models to generate counterparts in either direction.

Figure 3 shows human judgments of model-generated WME and model-generated AAL. With the exception of Flan-T5 (FT), we see that model-generated WME is judged as more human-like and closer to the intended language variety than model-generated AAL. These results confirm findings from automatic metrics, showing that models more easily generate WME than AAL. In contrast, for meaning and tone, the reverse is true, indicating that models generate WME that does not match the meaning of the original AAL. The difference between scores on AAL and WME were significant on all metrics for at least two of the models as determined by a two-tailed t-test of the means (see * in Figure 3 for models with significant differences). We see also that the drop in meaning and tone scores from judgments on human WME is larger than the drop in human-like and dialect scores on WME. These observations suggest that models have a hard time interpreting AAL.

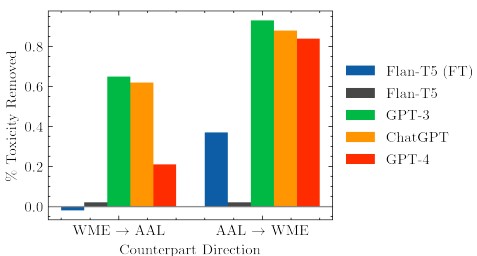

Figure 4: Percentage of toxicity removed for AAL and the respective aligned WME counterparts.

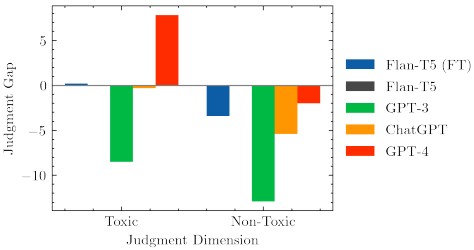

Figure 5: Gaps in Rouge-1 metric in non-toxic and toxic subsets of the AAL dataset. Negative Rouge gaps indicated greater WME performance than AAL.

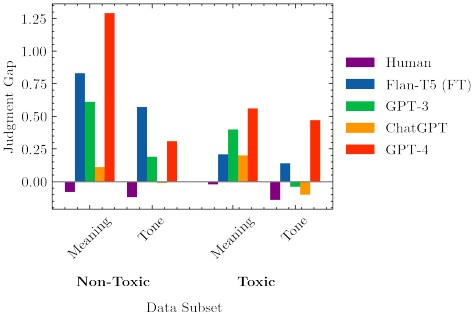

Figure 6: Gaps in human judgments of human and model-generated counterparts broken down into toxic and non-toxic subsets. Negative scores indicate better WME performance.

Toxicity scores (Figure 4) show that models tend to remove toxic words when generating both AAL and WME[10]. To test whether removal of toxic words contributed to the inability to preserve meaning, we computed both coverage scores and human evaluation scores on two subsets of the data: "toxic" texts (at least one term categorized as offensive) and "non-toxic" texts (no terms categorized as offensive). We display these results as the difference in coverage scores and in human judgment scores between model generated AAL and WME as shown in Figure 5 and Figure 6. Positive scores indicate that AAL performs better. Here we see that human judgments on meaning and tone show that generated WME is worse than generated AAL for both toxic and non-toxic subsets. Thus, differences in use of toxic words between input and output cannot be the sole cause for lower scores on meaning and tone. This confirms that models have difficulty interpreting features of AAL. We note furthermore

---

[10]Models are developed to avoid generating toxic or offensive language, so the trend of neutralizing input texts in any dialect is expected. There are notable differences, however, in the extent to which this neutralization occurs. The results show that a significantly higher proportion of toxic language is removed when generating WME from AAL than in the reverse direction.

that gaps in coverage are consistently larger for the non-toxic subsets of the data, demonstrating that the use of profanity and toxic language are also not the primary cause of gaps in coverage metrics.

### 5.3 Masked Span Prediction

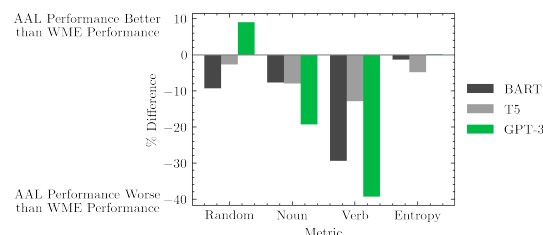

Figure 7: Percent difference in perplexity and top-5 entropy between AAL and aligned WME texts. Negative percentages indicate lower WME perplexity/entropy than AAL perplexity/entropy.

For MSP, we measure both perplexity and the entropy of generating a masked span in either AAL or WME. A lower perplexity score indicates it is easier for the model to determine the missing phrase, while a lower entropy score indicates the model places high probability in its top predictions. Figure 7 shows the differences in perplexity and entropy between masked span prediction for model-generated WME and for model-generated AAL.

Negative percent changes in perplexity indicate that it is easier for models to predict spans in WME than AAL, while for entropy, indicate that models place higher probability in their top predictions for WME than for AAL sentences.

## 6 Discussion: How well do models *interpret* AAL?

We discussed earlier how Figure 3 and Figure 6 demonstrate that models have difficulty interpreting AAL when generating WME. Figure 7 supports this finding as well, as models generally output higher perplexities for masked spans in AAL compared to aligned WME.

The largest gaps in perplexity between the two language varieties is assigned to masked verb phrases. One set of distinct features characterizing AAL are verbal aspects which do not occur in WME such as the future *gone* (e.g., *I'm gone do it later*), so this result may suggest that models struggle with the use of AAL-specific aspects in particular over other AAL features. A similar trend is found in the entropy metric, suggesting that AAL

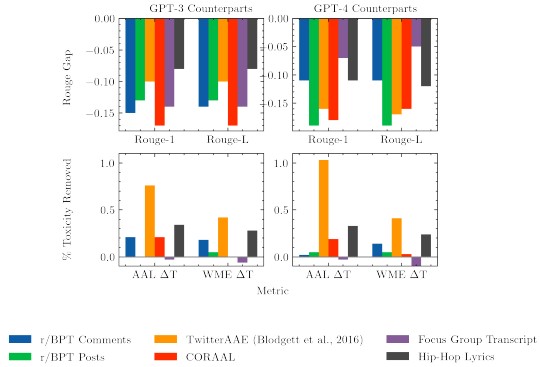

Figure 8: Breakdown of Rouge score gaps and percent changes in toxicity by data source in counterpart generation. Negative Rouge values indicate higher WME Rouge scores, and positive ΔT scores indicate the model-generated counterpart is more toxic than the gold standard.

| Ex 1: Source AAL | **shit, i'm tryna get my thing goin', too. just bring my car back. shit, he faded.** |
|---|---|
| **Annotated WME Text** | **Shit, I'm trying to do my own thing, too. Just bring my car back, shit, he's high.** |
| **Flan-T5 (FT) WME** | Hey, I'm trying to get my thing going, too. Just bring my car back. Hey, he faded. |
| **Flan-T5 (FT) AAL** | Shit, I'm trying to do my own thing, too. Just bring my car back, shit, he's high. |
| Ex 2: Source AAL | **been wilding since a juvi, she was a good girl.** |
| **Annotated WME** | **I was wild since I was a juvenile; she was a good girl.** |
| **ChatGPT WME** | I have been behaving recklessly since my time in juvenile detention, but she was a well-behaved girl. |
| **ChatGPT AAL** | I been wildin' since I was a shorty; she was a straight up good girl. |
| Ex 3: Source AAL | **It used to be broken controllers.** |
| **Annotated WME** | **There used to be broken controllers.** |
| **GPT-4 WME** | It used to be that controllers would break. |
| Ex 4: Source AAL | **and they ain't gonna really keep it one thousand when niggas- let a nigga know what the fuck really going on.** |
| **Annotated WME** | **And they are not really going to tell the truth when people- let me know what the fuck is really going on.** |
| **GPT-4 WME** | And they aren't going to be completely honest when people-inform someone about what is truly happening. |

Table 4: Flan-T5 (FT), ChatGPT and GPT4 examples of model counterparts neutralizing the input text (red) and misinterpreting features of AAL or terms in the input text (blue).

text also lowers model confidence in their own predictions. These results for AAL support similar findings by Groenwold et al. (2020) for GPT-2 in an auto-completion setting.

Manual inspection revealed more fine-grained patterns of model behavior within aspectual verbs and other AAL features. Models seem to correctly interpret some specific features of AAL, namely: the use of *ain't*, double negation, and habitual *be*.

Examples of misinterpretation, however, are shown in Table 4 illustrating difficulty with several other aspects of AAL. Several mistakes involve lexical interpretation, such as in example 1, where the model is not able to interpret the meaning of "he faded" as "he's high" and example 2 where the model inserts *shorty* apparently intending the mean-

ing "youth" instead of its more common meaning of "girlfriend". The models also struggle with features that appear the same as in WME, but have slightly different meanings in AAL. These include remote past *been* (example 2), which is incorrectly interpreted as past perfect (*have been*), and existential *it* (example 3), which in WME is closest in meaning to "there" as in "there are ..." and is not correctly interpreted by any model.

We also include an example where GPT-4 misinterprets the phrase "a nigga" as referencing another person, when in the provided context, the use most closely resembles referencing oneself. The word *nigga*, is one of the N-words, a set of lexical items well-documented by linguists as being misunderstood by non-native speaker in terms of their syntactic and semantic complexity (Rahman, 2012; Grieser, 2019; Smith, 2019). In particular, while the model removes the word in generating the WME counterpart, it does not correctly understand the use of the N-word as referencing the subject. Without this understanding, it is probable that models will both misinterpret the words as toxic and use them in ways that are considered offensive to the AAL-speaking community.

In additional analysis of counterpart generations, we examined model performance gaps in each subset of the AAL dataset. Among subsets of the data, gaps between Rouge-1 metrics for AAL and WME counterparts vary significantly. GPT-4, for example, presents the largest performance gap for the TwitterAAE corpus (Blodgett and O'Connor, 2017), and the smallest gaps for the hip-hop and focus group subsets as shown in Figure 8. Manual inspection reveals that this aligns with the trends in AAL-use among the subsets as well: distinct features of AAL appear to be more frequent in the TwitterAAE dataset, while AAL features in the focus group transcripts appear to be more sparse. This pattern may be due to the makeup and context of the focus groups, as most participants were college-educated, working professionals and selected to specifically describe grief experiences, possibly affecting the use of AAL in the discussions. These results may suggest, as would be expected, that higher density of AAL features leads to larger performance gaps.

## 7 Related Work

While few have specifically focused on bias against AAL in language generation, related work has ex-

tensively investigated societal biases in language tasks. One large-scale study (Ziems et al., 2022) investigates performance on the standard GLUE benchmark (Wang et al., 2018) using a synthetically constructed AAL dataset of GLUE for fine-tuning. They show performance drops on a small human-written AAL test set unless the Roberta model is fine-tuned.

**Racial Bias in Generation.** Mitigating and evaluating social biases in language generation models is a challenging problem due to the apparent trade-offs between task performance and bias mitigation, the many possible sources of bias, and the variety of biases and perspectives to examine (Sheng et al., 2021b; Akyürek et al., 2022). A number of studies have proposed bias evaluation measures, often using prompts crafted to reveal biased associations of, for example, occupation and gender (i.e., "The [man/woman] worked as a ...") (Sheng et al., 2020, 2019; Kiritchenko and Mohammad, 2018; Dhamala et al., 2021; Shen et al., 2022) and in other cases, graph representations to detect subjective bias in summarization (Li et al., 2021) and personas for dialogue generation (Sheng et al., 2021a). However, the bias measurements in many of these approaches are not directly applicable to language in a natural setting, where the real-life harmful impacts of bias in language generation would be more prevalent.

**AAL Feature Extraction.** Past work makes progress in lowering performance gaps between AAL and WME by focusing on linguistic feature extraction tasks. Given that some features of AAL such as the aspectual verbs (i.e., habitual *be*, remote past *been*) do not have equivalent meanings and functions in WME (Green, 2009), standard part-of-speech (POS) taggers and dependency parsers cannot maintain performance for AAL text. Studies have attempted to lessen this gap by creating a POS tagger specifically for AAL through domain adaptation (Jørgensen et al., 2016) and a dependency parser for AAL in Tweets (Blodgett et al., 2018). Beyond these tasks, considerable attention has been given to developing tools for features specific to AAL and other language varieties, such as detecting dialect-specific constructions (Masis et al., 2022; Demszky et al., 2021; Santiago et al., 2022; Johnson et al., 2022) to aid in bias mitigation strategies.

**AAL in Language Tasks.** Bias has also been measured specifically with respect to AAL in down-stream, user-facing tasks. With the phonological differences between AAL and WME, automatic speech recognition (ASR) systems have shown large performance drops when transcribing speech from African American speakers (Koenecke et al., 2020; Martin and Tang, 2020; Mengesha et al., 2021). Toxicity detection and offensive language classification models have also been evaluated and have shown a higher probability of incorrectly labeling AAL text as toxic or offensive when compared to WME text (Zhou et al., 2021; Rios, 2020; Sap et al., 2022). Most closely related to this work, one study evaluated bias against AAL in transformer generation models, showing that in a sentence auto-completion setting, GPT-2 generates AAL text with more negative sentiment than in aligned WME texts (Groenwold et al., 2020). Further investigation of both a larger set of language generation models as well as a broader set of generation tasks would provide a clearer picture of model biases against AAL.

## 8    Conclusion

We demonstrate through investigation of two tasks, counterpart generation and masked span prediction, that current LLMs have difficulty both generating and interpreting AAL. Our results show that LLMs do better matching the wording of gold standard references when generating WME than when generating AAL, as measured by Rouge and BERTScore. Human evaluation shows that LLM output is more likely to be judged as human-like and to match the input dialect when generating WME than AAL. Notably, however, LLMs show difficulty in generating WME that matches the meaning and tone of the gold standard, indicating difficulty in interpreting AAL. Our results suggest that more work is needed in order to develop LLMs that can appropriately interact with and understand AAL speakers, a capability that is important as LLMs are increasingly deployed in socially impactful contexts (e.g., medical, crisis).

## Limitations

We acknowledge a few limitations accompanying the evaluation of biases in LLMs. While our analysis is primarily restricted to intrinsic evaluation of model biases, users primarily interact with LLMs in a chat-based interface such as with ChatGPT, or use the model for specific tasks such as question answering. This approach was chosen to analyze

biases that would be present across all tasks involving AAL. Performance gaps and biases analyzed in a task-specific setting, however, may yield different trends than presented in this paper, and we leave this investigation to future work.

Additionally, AAL exhibits significant variation by region, context, speaker characteristics, and many other variables. We attempt to more comprehensively reflect real AAL use by drawing text from multiple sources and contexts, but are ultimately limited by the data available. For example, while CORAAL reflects natural AAL speech, it is limited to a select set of regions (e.g., New York, Georgia, North Carolina, Washington DC), and while the Twitter and Reddit AAL subsets may span many regions, they are also influenced by the linguistic features of social media. Similar biases may also exist in other underrepresented varieties of English such as Mexican American English, Indian English, or Appalachian English. Due to the availability of data, we focus on AAL, but given texts in other varieties, this work could be extended to examine biases regarding these and other language varieties.

Finally, evaluation metrics relying on trained models or lexicons, such as BERTScore and toxicity measures, may also inherently encode biases concerning AAL text. Rather than using a model to measure toxicity, we instead use a lexicon of offensive terms provided in Zhou et al. (2021) and used to measure lexical biases in toxicity models. Given that analyzing performance gaps relies on accurate and unbiased measures of model performance, future work may give attention to developing unbiased language generation metrics.

## Ethics Statement

We recognize that identifying potential bias against AAL in LLMs should also include a critically reflexive analysis of the consequences if language models are better at understanding language varieties specific to marginalized communities such as AAL, and the extent to which that impacts those speakers. In prior research, Patton et al. (2020) have noted that decisions made by researchers engaged in qualitative analysis of data through language processing should understand the context of the data and how algorithmic systems will transform behavior for individual, community, and system-level audiences. Critical Race Theory posits that racism exists across language

practices and interactions (Delgado and Stefancic, 2023). Without these considerations, LLMs capable of understanding AAL could inadvertently be harmful in contexts where African Americans continue to be surveilled (e.g., social media analysis for policing).

Despite this, including African American representation in language models could potentially benefit AAL speakers in socially impactful areas, such as mental health and healthcare (e.g., patient notes that fully present the pain African American patients are experiencing in the emergency room, Booker et al. 2015). Considering both the potential for misuse of the data as well as the potential for social good, we will make the Tweet IDs and other collected data available to those that have signed an MOU indicating that they will use the data for research purposes only, limiting research to improved interpretation of AAL in the context of an application for social good. In the MOU, applicants must include their intended use of the data and sign an ethics agreement.

## Acknowledgements

This work was supported in part by grant IIS-2106666 from the National Science Foundation, National Science Foundation Graduate Research Fellowship DGE-2036197, the Columbia University Provost Diversity Fellowship, and the Columbia School of Engineering and Applied Sciences Presidential Fellowship. Any opinion, findings, and conclusions or recommendations expressed in this material are those of the authors and do not necessarily reflect the views of the National Science Foundation. We thank the anonymous reviewers and the following people for providing feedback on an earlier draft: Tuhin Chakrabarty, Esin Durmus, Fei-Tzin Lee, Smaranda Muresan, and Melanie Subbiah. We also thank Mayowa Fageyinbo, Gideon Kortenhoven, Kendall Lowe, and Tajh Martin for providing annotations.

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

# A   Data Statement

We provide details about our dataset in the following data statement. Much of the dataset is drawn from existing datasets that lack data statements, and in those cases, we include what information we can.

## A.1   Curation Rationale

The dataset was collected in order to study the robustness of LLMs to features of AAL. The data is composed of AAL-usage in a variety of regions and contexts to capture the variation in the use of and density of features. In order to better ensure the included texts reflect AAL, we sample texts from social media, sociolinguistic interviews, focus groups, and hip-hop lyrics and weight the probability of sampling a text using a small set of known AAL morphosyntactic features. The datasets that were previously collected, CORAAL Kendall and Farrington 2021 and TwitterAAE (Blodgett et al., 2016), were originally created to study AAL and to study variation in AAL on social media respectively. For all texts in the dataset, we also collect human-annotated counterparts in WME to provide a baseline for model evaluations.

## A.2   Language Variety

All texts included in the dataset are in English (en-US) as spoken or written by African Americans in the United States with a majority of texts reflecting linguistic features of AAL. Some texts notably contain no features of AAL and reflect WME.

## A.3   Speaker Demographics

Most speakers included in the dataset are African American. The r/BPT texts were restricted to users who have been verified as African American, CORAAL and focus group transcripts were originally interviews with African Americans, and hip-hop lyrics were restricted to African American artists. The TwitterAAE dataset is not guaranteed to be entirely African American speakers, but the texts are primarily aligned with AAL and have a high probability of being produced by AAL speakers. Other demographics such as age and gender are unknown.

## A.4   Annotator Demographics

While all AAL texts in the dataset reflect natural usage of AAL, the WME counterparts in the dataset are annotated. We recruited 4 human annotators to generate WME counterparts for each text. All annotators self-identify as African American, self identify as AAL speakers, and are native English speakers. Additionally, the 4 annotators are undergraduate and graduate students aged 20-28, 2 of whom were graduate students in sociolinguistics. All annotators were compensated at a rate between $18 and $27 per hour depending the annotator's university and whether they were an undergraduate or graduate student.

## A.5   Speech Situation

Speech situations vary among the 6 datasets we compose. The r/BPT posts, r/BPT comments,

and TwitterAAE subsets are all originally type-written text, intended for a broad audience, and are drawn from asynchronous online interactions. The CORAAL and focus group transcript subsets are originally spoken and later transcribed, intended for others in their respective conversations, and are drawn from synchronous in-person interactions. Finally, the hip-hop lyrics subset are both spoken and written, intended for a broad audience of hip-hop listeners, and are likely repeatedly changed and edited before released. r/BPT comments and posts are sampled from the origin of the subreddit in October 2015, CORAAL transcripts are sampled from interviews between 1888 and 2005, hip-hop lyrics are drawn from songs released in 2022, focus groups were conducted between February and November 2022, and the time range of the Twitter-AAE dataset is unknown to the authors.

## A.6 Text Characteristics

Among the data subsets, the focus group transcripts are the most topically focused. All focus groups primarily included discussion surrounding the experiences and responses to grief in the Harlem community, focusing on experiences due to daily stressors, the death of loved ones, police shootings, and the COVID-19 pandemic. In the r/BPT posts and r/BPT comments subsets, texts were typically written in response to a tweet by an African American Twitter user, ranging from political commentary to discussion of the experience of African Americans in the United States. The hip-hop lyrics subset is not topically focused, but includes texts that follow specific rhyming patterns and meters. The remaining subsets of the data (TwitterAAE, CORAAL) span a variety of topics and structures.

## B  AAL Search Patterns

To better ensure our dataset includes use of AAL features, we use a set of regex and grammar-based search patterns as part of the sampling procedure. Regex patterns for AAL features are listed below.

| AAL Feature | Pattern |
|---|---|
| *ain't* | ain'?t |
| Existential *it* | it (?:was\|is) a\w* |
| Negative Concord | n't (no\w*)\|n't (?:nobody\|anybody) |
| Dropped Copula | (\bthey\|\bwe\|\bshe\|\bhe) \w*?ing \b |
| Determiner Leveling | a [aeiou]\w+ |

Table 5: ALL feature search patterns.

The set also includes grammar-based patterns using the spacy POS tagger to detect the use of habitual *be*, completive *done* (or "dun", "dne"), future

*gone* (or "gne", "gon"), and remote past *been* (or "bin"). Each of these features are detected by the use of each term (or their variants) if they are not preceded by another auxiliary verb or preposition in the clause (i.e., "He be eating" contains a use of habitual *be*, but "Should he be eating?" does not because the auxiliary verb "should" precedes "be" in the clause). While standard POS taggers could potentially underperform on AAL, there were no AAL-specific POS taggers available at the time of dataset collection to our knowledge.

## C  Annotation Procedure

### C.1  Counterpart Annotations

Annotators were asked to provide a semantically-equivalent rewriting (or *counterpart*) of a given text from the AAL dataset in WME. The specific set of guidelines provided to annotators were:

1. Change alternative spellings (i.e., "shoulda" for "should've")
2. Maintain usernames, hashtags, and URLs if present
3. Ignore emojis unless speakers of WME may use them differently
4. Conserve and un-censor profanity
5. Avoid unnecessary changes
6. Use your best judgement in special cases

As noted, annotators had the option to label a text as "Not Interpretable" if it lacks a reasonable counterpart in WME.

### C.2  Counterpart Judgment Instructions

In judging counterparts, annotators were provided with an original text from the dataset in either WME or AAL and a model-generated (or human-annotated) counterpart. Notably, judgments were assigned ensuring that no annotator received a text they were involved in creating the counterpart for. Additionally, annotators were not given definitions or guidelines for terms such as "meaning" or "tone" to avoid biasing judgements and to encourage annotators to use their own interpretation of the terms. The questions asked of annotators are as follows:

1. *Human-likeness*: Is the interpretation more likely generated by a human or language model?
   **(1)** Very Likely a Model, **(2)** Likely a Model, **(3)** Neutral, **(4)** Likely a Human, **(5)** Very Likely a Human

2. *Linguistic Match*: How well does the interpretation resemble AAL?
   **(1)** Completely Unlike AAL, **(2)** Unlike AAL, **(3)** Neutral, **(4)** Like AAL, **(5)** Completely Like AAL
3. *Meaning Preservation*: How well does the interpretation reflect the meaning of the original text?
   **(1)** Completely innacurate, **(2)** Inaccurate, **(3)** Neutral, **(4)** Accurate, **(5)** Completely Accurate
4. *Tone Preservation*: How well does the counterpart accurately reflect the tone of the original text?
   **(1)** Completely innacurate, **(2)** Inaccurate, **(3)** Neutral, **(4)** Accurate, **(5)** Completely Accurate

For judgment samples that involved judging WME counterparts, questions and response options that refer to "AAL" were changed accordingly.

## D Annotator Agreement Calculation

Because the task provided to annotators required generating text, we use Levenshtein distance and Krippendorf's Alpha to calculate annotator agreement based on the general form as described in Braylan et al. 2022. Annotator agreement is calculated with the following formula:

$$\alpha = 1 - \frac{\hat{D}_o}{\hat{D}_e} \qquad (1)$$

where $\hat{D}_o$ represents the observed distance between annotations, $\hat{D}_e$ represents the expected distance, and Levenshtein distance is used as the distance function $D(a, b)$. Expected distance between two annotators is calculated by randomly shuffling one set of annotations and calculating the average Levenshtein distance between the randomized pairs.

## E Model and Experiment Details

### E.1 Checkpoints and Training

For GPT-3, ChatGPT, and GPT-4, we use the *text-davinci-003*, *gpt-3.5-turbo*, and *gpt-4* checkpoints respectively . For the T5 model variants, we use the *t5-large* and *google/flan-t5-large* checkpoint. Flan-T5 is fine-tuned using a learning rate of 3e-5 for 5 epochs across the full set of 72 additional texts. Finally, for BART we use the *facebook/bart-large* checkpoint.

### E.2 Generation Hyperparameters

For all GPT-family models, we use the default temperature of 0.7 in generations. For the BART and T5 model variants, we use a beam width of 3, temperature of 1, and a *no_repeat_ngram_size* of 3.

## F Full Counterpart Results

Table 6 and Table 7 show the raw automatic metric and human judgement scores for the counterpart generation task respectively. Table 8 shows the percentage of toxic terms removed by the model when generating counterparts in AAL or WME. Finally, Table 9 and Table 10 present the raw automatic metric and human judgments scores on the toxic (at least one term categorized as offensive) and non-toxic (no terms categorized as offensive) subsets of the corpus.

## G Full MSP Results

Table 11 presents the raw perplexity and entropy scores in the Masked Span Prediction task.

## H Additional Counterpart Generation Examples

The remaining appendices, Tables 12-19, provide further examples from the counterpart generation task. Examples are drawn randomly from subsets where the total score given to one of the models evaluated exceeds the ratings of the original annotated counterpart and where the total score of a model is lower.

| | AAL ⇔WME | | | | | | | | | | |
|---|---|---|---|---|---|---|---|---|---|---|---|
| Model | WME → AAL | | | | | | AAL → WME | | | | |
| | R-1 | R-2 | R-L | BS | T | %Δ T | R-1 | R-2 | R-L | BS | T | %Δ T. |
| *Repeat* | 77.4 | 59.8 | 75.7 | 53.1 | 0.25 | — | 77.4 | 59.8 | 75.7 | 53.1 | 0.33 | — |
| Flan-T5 (FT) | 77.2 | 63.6 | 76.1 | 53.5 | 0.21 | -0.02 | 80.3 | 67.8 | 79.6 | 54.4 | 0.26 | 0.38 |
| Flan-T5 | 74.8 | 57.7 | 74.3 | 52.4 | 0.30 | 0.01 | 75.8 | 58.6 | 75.3 | 53.9 | 0.23 | 0.02 |
| GPT-3 | 56.5 | 32.9 | 55.2 | 51.6 | 0.02 | 0.60 | 69.5 | 51.0 | 68.2 | 53.5 | 0.13 | 0.92 |
| ChatGPT | 56.2 | 32.7 | 54.0 | 51.2 | 0.05 | 0.47 | 61.6 | 40.2 | 59.3 | 53.8 | 0.15 | 0.90 |
| GPT-4 | 67.7 | 45.6 | 66.9 | 51.7 | 0.06 | 0.16 | 68.9 | 48.9 | 67.5 | 53.7 | 0.21 | 0.84 |

Table 6: Full coverage and toxicity metrics of model counterpart generations for AAL and aligned WME. Toxicity scores are reported both as raw scores (T) and as the percent change in scores (%ΔT) between model inputs to corresponding model-generated counterparts. *Repeat* row reports metrics computed on and between gold standard reference texts.

| | AAL ⇔WME | | | | | | | |
|---|---|---|---|---|---|---|---|---|
| **Model** | **Human-like** | | **Dialect** | | **Meaning** | | **Tone** | |
| | AAL | WME | AAL | WME | AAL | WME | AAL | WME |
| *Human* | 3.66 | 3.82 | 4.00 | 4.99 | 4.51 | 4.53 | 4.55 | 4.67 |
| Flan-T5 (FT) | **4.01** | 3.56 | 3.80 | 3.61 | **4.79** | 4.45 | **4.83** | 4.61 |
| GPT-3 | 2.65 | 3.50 | 3.56 | 4.62 | 4.18 | 3.72 | 4.07 | 4.04 |
| ChatGPT | 3.18 | 3.62 | **4.09** | 4.67 | 3.99 | 3.80 | 4.11 | 4.19 |
| GPT-4 | 3.17 | 3.51 | **4.05** | 4.74 | **4.67** | 3.96 | **4.65** | 4.21 |

Table 7: Full ratings of counterpart responses for Flan-T5, GPT-3, ChatGPT, GPT-4, and the original human response in AAL and WME. Bolded values indicate scores exceeding judgments of human counterparts. The higher score between AAL and WME is colored green.

| **Model** | **% Toxicity Removed** | |
|---|---|---|
| | WME → AAL | AAL → WME |
| Flan-T5 (FT) | -0.02 | 0.37 |
| Flan-T5 | 0.01 | 0.02 |
| GPT-3 | 0.65 | 0.93 |
| ChatGPT | 0.62 | 0.88 |
| GPT-4 | 0.21 | 0.84 |

Table 8: Percentage of toxicity removed from model inputs when generating counterparts in the WME → AAL and AAL → WME directions. Negative values indicate more toxic terms were introduced in model-generated counterparts than were present in the input.

| **Model** | **Toxic** | | | **Non-Toxic** | | |
|---|---|---|---|---|---|---|
| | AAL | WME | Difference | AAL | WME | Difference |
| Flan-T5 (FT) | 74.9 | 74.7 | 0.2 | 78.9 | 82.3 | -3.4 |
| Flan-T5 | 70.3 | 69.0 | -0.01 | 75.6 | 77.5 | 0.02 |
| GPT-3 | 54.0 | 62.5 | -8.5 | 56.5 | 69.4 | -12.9 |
| ChatGPT | 56.7 | 57.0 | -0.3 | 57.5 | 62.9 | -5.4 |
| GPT-4 | 68.2 | 60.4 | 7.8 | 65.9 | 67.9 | -2.0 |

Table 9: Rouge-1 gaps in Toxic and Non-Toxic subsets of the AAL dataset. Negative gaps indicate greater WME performance than AAL performance.

| **Model** | **Toxic** | | | | **Non-Toxic** | | | |
|---|---|---|---|---|---|---|---|---|
| | Human-like | Dialect | Meaning | Tone | Human-like | Dialect | Meaning | Tone |
| Human | **-0.70** | 0.40 | **-0.08** | -0.12 | -0.06 | **-0.10** | -0.02 | **-0.14** |
| Flan-T5 (FT) | 0.45 | 0.49 | 0.83 | 0.57 | **0.44** | **0.11** | **0.21** | **0.14** |
| GPT-3 | **-0.82** | -1.04 | 0.61 | 0.19 | -0.88 | **-1.07** | 0.40 | **-0.04** |
| ChatGPT | -0.51 | **-0.60** | **0.11** | -0.01 | **-0.42** | -0.57 | 0.20 | **-0.10** |
| GPT-4 | 0.33 | -0.31 | 1.29 | 0.31 | **-0.50** | **-0.7** | **0.56** | 0.47 |

Table 10: Gaps in human judgments of counterparts in the toxic subset and non-toxic subset of the data. Negative scores indicate higher WME performance, and bolded scores indicate smaller gaps (less positive or more negative) compared to the other subset.

| AAL | | | | | | |
|---|---|---|---|---|---|---|
| **Model** | **Perplexity** | | | **Top-5 Entropy** | | |
| | Random | Noun | Verb | Random | Noun | Verb |
| GPT-3 | 8.0 | 21.0 | 12.2 | 12.24 | 15.86 | 13.80 |
| T5 | 1737.79 | 3657.64 | 2420.74 | 1.53 | 1.51 | 1.50 |
| BART | 2559.9 | 5856.9 | 3900.5 | 5.56 | 5.99 | 5.67 |
| WME | | | | | | |
| **Model** | **Perplexity** | | | **Top-5 Entropy** | | |
| | Random | Noun | Verb | Random | Noun | Verb |
| GPT-3 | 8.79 | 17.6 | 8.76 | 13.40 | 15.32 | 13.19 |
| T5 | 1691.58 | 3391.09 | 2144.63 | 1.26 | 1.33 | 1.31 |
| BART | 2342.71 | 5437.39 | 3015.38 | 5.56 | 5.94 | 5.52 |

Table 11: Span perplexity and top-5 entropy for all AAL and aligned WME texts. Results for T5 and BART are averaged across 5 trials with different random spans masked.

| Source | Text | Human-like | Dialect | Meaning | Tone |
|---|---|---|---|---|---|
| **Human WME Counterpart** | **Somebody is going to get robbed for that new money. Just watch because this niggas are broke and desperate** | - | - | - | - |
| **AAL Text** | **Somebody going get bust for them new breads watch cause niggas be broke n press.** | 1.0 | 2.0 | 1.0 | 1.0 |
| Flan-T5 (FT) Generated AAL | Somebody gonna get robbed for that new money. Just watch because these niggas are broke and desperate. | 5.0 | 4.0 | 5.0 | 5.0 |
| GPT-3 Generated AAL | Somebody gon' get robbed fo' dat new money. Just watch 'cause these niggas po' an' 'esperate. | 3.0 | 3.0 | 4.5 | 4.5 |
| ChatGPT Generated AAL | Somebody finna get jacked for that new cash. Imma tell ya, cause these niggas out here strugglin' and needin' it bad. | 4.5 | 4.5 | 4.5 | 5.0 |
| GPT-4 Generated AAL | Somebody gon' get got for that new money. Just watch cuz these niggas broke and desperate. | 3.0 | 5.0 | 5.0 | 5.0 |

Table 12

| Source | Text | Human-like | Dialect | Meaning | Tone |
|---|---|---|---|---|---|
| **Human WME Counterpart** | **And what if he does? People are going to be powerless to stop it.** | - | - | - | - |
| **AAL Text** | **And what if he does? Niggas gon be powerless to stop that shit too.** | 1.0 | 3.0 | 2.0 | 2.0 |
| Flan-T5 (FT) Generated AAL | And what if he does? People gonna be powerless to stop it. | 4.5 | 5.0 | 5.0 | 5.0 |
| GPT-3 Generated AAL | An' wat if he do? Folks gon' be powerless ta stop it. | 1.5 | 3.0 | 5.0 | 4.5 |
| ChatGPT Generated AAL | And what if he do though? Ain't nobody gonna have no power to stop it. | 4.0 | 5.0 | 5.0 | 5.0 |
| GPT-4 Generated AAL | An' what if he do? Folks gon' be powerless to stop it. | 3.5 | 5.0 | 5.0 | 5.0 |

Table 13

| Source | Text | Human-like | Dialect | Meaning | Tone |
|---|---|---|---|---|---|
| **Human WME Counterpart** | **Number one top is really a number one thing that causes everything I know all the violence, he would just go back and forth and social media.** | - | - | - | - |
| **AAL Text** | **Number one top really a number one thing that causes everything I know all the violence, he would just go back and forth and social media.** | 5.0 | 5.0 | 5.0 | 5.0 |
| Flan-T5 (FT) Generated AAL | Number one top is really a number one thing that causes everything I know all the violence, he would just go back and forth and social media. | 5.0 | 5.0 | 5.0 | 5.0 |
| GPT-3 Generated AAL | Num one top is suttin' that's really low-key cuz it be causin' all da violence I know. He just keep goin' back and forth on da social media, know'msayin? | 1.0 | 1.0 | 4.0 | 4.0 |
| ChatGPT Generated AAL | Da numba one top thang be causin' errythang, ya feel me? I know all da violence be 'cause of it. Dude jus' be goin' back and forth on social media or whatever. | 1.0 | 5.0 | 1.0 | 2.0 |
| GPT-4 Generated AAL | Numba one top be straight up da numba one thing causin' all dis violence I know, he just be goin' back and forth on social media n' stuff. | 4.0 | 3.0 | 5.0 | 5.0 |

Table 14

| Source | Text | Human-like | Dialect | Meaning | Tone |
|---|---|---|---|---|---|
| **Human WME Counterpart** | **My man, i diss in my freestyle rhyme** | - | - | - | - |
| **AAL Text** | **my man, i be dissin' in my freestyle rhyme.** | 4.5 | 4.5 | 4.5 | 4.5 |
| Flan-T5 (FT) Generated AAL | my man, i diss in my freestyle rhyme | 5.0 | 5.0 | 5.0 | 5.0 |
| GPT-3 Generated AAL | My dude, I was killin' it in my freestyle rhyme! | 1.0 | 2.0 | 2.0 | 1.0 |
| ChatGPT Generated AAL | My homie, I dissed in my freestyle rap. | 2.5 | 3.0 | 4.0 | 4.5 |
| GPT-4 Generated AAL | My dude, I be dissin' in my freestyle flow | 2.0 | 4.0 | 4.0 | 5.0 |

Table 15

| Source | Text | Human-like | Dialect | Meaning | Tone |
|---|---|---|---|---|---|
| **AAL Text** | **now they sayin' juice left some heads cracked.** | - | - | - | - |
| **Human WME Counterpart** | **Now they're saying that juice left some heads cracked.** | 1.5 | 2.0 | 1.5 | 2.5 |
| Flan-T5 (FT) Generated WME | Now they say juice left some heads cracked. | 3.5 | 2.0 | 5.0 | 5.0 |
| GPT-3 Generated WME | Now they are saying that Juice left some people with serious injuries. | 5.0 | 5.0 | 4.0 | 5.0 |
| ChatGPT Generated WME | Now they are saying that Juice caused some people to have head injuries. | 2.0 | 4.0 | 5.0 | 5.0 |
| GPT-4 Generated WME | Now they are saying that Juice injured some people. | 5.0 | 5.0 | 4.0 | 4.0 |

Table 16

| Source | Text | Human-like | Dialect | Meaning | Tone |
|---|---|---|---|---|---|
| **AAL Text** | **Moms always buyin groceries like they preparing for a nuclear winter.** | - | - | - | - |
| **Human WME Counterpart** | **Mam is always buying groceries like they are preparing for a nuclear winter.** | 2.0 | 2.0 | 2.0 | 5.0 |
| Flan-T5 (FT) Generated WME | Moms always buy groceries like they are preparing for a nuclear winter. | 3.5 | 4.5 | 5.0 | 5.0 |
| GPT-3 Generated WME | Mom always buys groceries like she's preparing for a nuclear winter. | 5.0 | 5.0 | 4.0 | 4.0 |
| ChatGPT Generated WME | My mother always buys groceries as if she is preparing for a nuclear winter. | 4.0 | 4.0 | 2.0 | 5.0 |
| GPT-4 Generated WME | Mom is always buying groceries as if she is preparing for a nuclear winter. | 5.0 | 5.0 | 2.0 | 2.0 |

Table 17

| Source | Text | Human-like | Dialect | Meaning | Tone |
|---|---|---|---|---|---|
| **AAL Text** | **It used to be broken controllers.** | - | - | - | - |
| **Human WME Counterpart** | **There used to be broken controllers.** | 5.0 | 5.0 | 4.5 | 4.5 |
| Flan-T5 (FT) Generated WME | It used to be broken controllers. | 3.0 | 1.5 | 1.5 | 3.5 |
| GPT-3 Generated WME | It used to be malfunctioning controllers. | 1.5 | 5.0 | 4.5 | 5.0 |
| ChatGPT Generated WME | It used to be controllers that were broken. | 1.5 | 5.0 | 2.5 | 3.5 |
| GPT-4 Generated WME | It used to be that controllers would break. | 2.5 | 5.0 | 1.5 | 1.5 |

Table 18

| Source | Text | Human-like | Dialect | Meaning | Tone |
|---|---|---|---|---|---|
| **AAL Text** | **that shit be crazy, they 'posed to protect us.** | - | - | - | - |
| **Human WME Counterpart** | **That shit is often crazy, they supposed to protect us.** | 3.0 | 4.0 | 4.5 | 4.5 |
| Flan-T5 (FT) Generated WME | That's crazy, they're trying to protect us. | 2.0 | 4.0 | 1.0 | 1.0 |
| GPT-3 Generated WME | That stuff is really outrageous, they're supposed to protect us. | 4.0 | 5.0 | 3.0 | 5.0 |
| ChatGPT Generated WME | That is insane, they are supposed to protect us. | 5.0 | 4.5 | 4.5 | 4.5 |
| GPT-4 Generated WME | That situation is crazy, they are supposed to protect us. | 4.5 | 4.5 | 4.5 | 4.5 |

Table 19