# OpenReview forum: "Evaluation of African American Language Bias in Natural Language Generation"
_EMNLP/2023/Conference — EMNLP 2023 Main_

### Official Review · Reviewer_jtk5 · 2023-08-03

**Soundness:** 4

**Excitement:**

5: Transformative: This paper is likely to change its subfield or computational linguistics broadly. It should be considered for a best paper award. This paper changes the current understanding of some phenomenon, shows a widely held practice to be erroneous in someway, enables a promising direction of research for a (broad or narrow) topic, or creates an exciting new technique.

**Paper Topic And Main Contributions:**

This paper conducts an in-depth investigation into the understanding of African-American English (AAE) by large language models (LLMs) in comparison to Standard American English (SAE). The authors assess a variety of LLMs, both fine-tuned and few-shot, on two tasks: (1) conversion between American English variants (e.g., AAE to SAE or vice versa), and (2) masked span prediction for AAE.

In the first task, the authors highlight the role of perceived toxicity in LLMs' understanding of AAE. They find that LLMs are more likely to infer toxicity in AAE text, possibly due to the unfamiliar context within SAE. This finding is significant as it sheds light on potential biases in LLMs.

In the second task, the authors demonstrate that LLMs have a poorer understanding of AAE compared to SAE. This is a crucial finding, as it underscores the need for more inclusive training data for LLMs.

Additionally, the authors contribute a dataset of AAE text samples, which is a valuable resource for future research in this area.

**Reasons To Accept:**

The paper provides a comprehensive and rigorous evaluation of LLMs' understanding of AAE through well-designed tasks. The experimental design tests multiple LLMs, including GPT-3, GPT-3.5 (ChatGPT), GPT-4, T5, Flan-T5, and BART, encompassing both decoder-only and encoder-decoder transformer-based LLMs.

The authors also experiment with different tuning paradigms: few-shot and fine-tuning. This approach provides a broad perspective on the performance of LLMs under different conditions.

The methodology, experimental design, and results are clear, reasonable, and easy to follow. The authors' findings are well-supported by the data and contribute to our understanding of LLMs' performance on different English variants.

The creation of an AAE text sample dataset is a significant contribution that can facilitate further research in this area.

Despite the predictable conclusion that LLMs perform better on English variants better represented in their training data, this paper provides empirical data supporting this for AAE, which is a valuable contribution to the field.

**Reasons To Reject:**

The paper would benefit from information about the demographics of the annotators and human evaluators. The authors demonstrate that LLMs perform worse on AAE-related tasks, but if these models are trained on predominantly SAE, individuals more familiar with SAE or AAE might misinterpret the samples.

**Reproducibility:**

4: Could mostly reproduce the results, but there may be some variation because of sample variance or minor variations in their interpretation of the protocol or method.

**Reviewer Confidence:**

4: Quite sure. I tried to check the important points carefully. It's unlikely, though conceivable, that I missed something that should affect my ratings.

**Typos Grammar Style And Presentation Improvements:**

For clarity, the authors could maintain the terminology used in previous literature. For instance, the abbreviation AAE could be used instead of AAL. However, the authors could still make it clear that they disagree with the current terminology.

---

> ### Author Rebuttal · Authors · 2023-08-29
>
> Thank you for your review.
>
> __Reason to Reject 1) The paper would benefit from information about the demographics of the annotators and human evaluators. The authors demonstrate that LLMs perform worse on AAE-related tasks, but if these models are trained on predominantly SAE, individuals more familiar with SAE or AAE might misinterpret the samples.__
>
> All annotators are African American undergraduate and graduate students, native AAL-speakers, and fluent WME-speakers. Given that certain cases of AAL can be extremely difficult for White Americans to understand, we elect to only choose annotators that are native AAL-speakers given they are able to better understand both dialects. We will update the discussion of both annotations and human judgments to explicitly note the race of annotators.
>
> __Typos, Grammar, Style And Presentation Improvement 1) For clarity, the authors could maintain the terminology used in previous literature. For instance, the abbreviation AAE could be used instead of AAL. However, the authors could still make it clear that they disagree with the current terminology.__
>
> The terms we use, “African American Language” and “White Mainstream English”, are both found in recent sociolinguistics literature (Alim & Smitherman, 2012; Baker-Bell, 2020). The authors that introduced these terms did so deliberately to “foreground the relationship between language, race, anti-Black racism, and white linguistic supremacy” (Baker-Bell, 2020), and so we adopt this terminology to recognize the social context of the work. We will update the Introduction to make clear that both terms are derived from prior sociolinguistics literature.
>
> H Samy Alim and Geneva Smitherman. 2012. Articulate while Black: Barack Obama, language, and race in the US. Oxford University Press.
>
> April Baker-Bell. 2020. Linguistic justice: Black language, literacy, identity, and pedagogy. Routledge.

---

### Official Review · Reviewer_gaMz · 2023-08-04

**Soundness:** 1

**Excitement:**

2: Mediocre: This paper makes marginal contributions (vs non-contemporaneous work), so I would rather not see it in the conference.

**Paper Topic And Main Contributions:**

This paper analyzes the performance of LLMs for African American Language. A dataset was created and experiments were run for two tasks (the counterpart generation task and the masked span prediction task).

**Questions For The Authors:**

In Figure 2, all the LLMs have comparable performance to Human in BERTScore. Does it mean LLMs are not worse than Human?

**Reasons To Accept:**

A dataset for African American Language was made, though the size is small (346 samples).

**Reasons To Reject:**

- The setting of the experiment is not valid.

In Section 4.1, the authors are comparing the differences of the performance for WME-to-AAL and AAL-to-WME. This experiment is not valid since we cannot compare results obtained from different test data (e.g., comparing translation results of English-to-German and German-to-English).

- The motivation and importance of the study is not clear.

There are various dialects and minority languages. This paper considers African American Language, but it is not clear why the specific language is handled here. The paper may be more appropriate for a focused workshop.
In this paper, "bias" is defined as "gaps in performance between one language and the other language" (L108). It is not clear why such differences are considered as bias, and it can be misleading.
This paper considers two simple tasks, but it is unclear if any social impacts can be estimated from the results.

**Reproducibility:**

3: Could reproduce the results with some difficulty. The settings of parameters are underspecified or subjectively determined; the training/evaluation data are not widely available.

**Reviewer Confidence:**

3: Pretty sure, but there's a chance I missed something. Although I have a good feel for this area in general, I did not carefully check the paper's details, e.g., the math, experimental design, or novelty.

**Typos Grammar Style And Presentation Improvements:**

In the paper, expressions like "model's interpretation" and "model's understanding" are often used (e.g., L232). They are somewhat confusing. The statistical models just solve optimization problems for given tasks.

---

> ### Author Rebuttal · Authors · 2023-08-29
>
> Thank you for your review.
>
> __Reason to Reject 1) The setting of the experiment is not valid. In Section 4.1, the authors are comparing the differences of the performance for WME-to-AAL and AAL-to-WME. This experiment is not valid since we cannot compare results obtained from different test data (e.g., comparing translation results of English-to-German and German-to-English).__
>
> Within translation literature, we found that it is common practice to make claims based on a comparison of model performance across test sets in different languages (e.g. Belinkov et al., 2017; Macketanz et al., 2021; Koehn, 2005). We acknowledge that there is also work arguing against such comparison, namely, a study by Bugliarello et al. (2020) which cites two reasons that n-gram overlap metrics cannot be compared cross-lingually: (1) different languages may use drastically different numbers of words to convey the same meaning, and (2) different languages utilize different tokenization schemes.
>
> We note first that while our task is framed as a translation task, the dialects we study are not distinct languages, which is why we use the name “counterpart generation” instead. Given the two criteria above, our task also avoids both of the weaknesses identified by Bugliarello et al. (2020). First, the AAL and WME texts in our dataset have a total of 5632 and 5665 words respectively, and aligned text samples differ by 0.095 words on average, making the first weakness of cross-lingual comparison unlikely to apply. Additionally, Bugliarello et al. (2020) finds that among linguistic properties, inherent difficulty of translating from a source language is most and negatively correlated with Type-Token Ratio (TTR). We have calculated the TTR for both the AAL (0.274) and WME (0.260) texts and find that there are no significant differences (p > 0.05) between the dialects in our dataset. Finally, because the comparisons in our experiments use the same models, they also use the same tokenization schemes, so the second weakness also does not apply. Therefore, we believe that the setting of the counterpart generation experiment and our comparisons using n-gram overlap metrics are valid.
>
> Furthermore, the counterpart generation task only represents a portion of our experiments and analyses, and thus does not impact the contributions of the entire paper. The conclusions resulting from this task are supported by the results of the masked span prediction task as well as qualitative analysis of models’ outputs. We will add a new section in the Appendix to include the above new analysis and discussion.
>
> Yonatan Belinkov, Nadir Durrani, Fahim Dalvi, Hassan Sajjad, and James Glass. 2017. What do neural machine translation models learn about morphology? In Proceedings of the 55th Annual Meeting of the Association for Computational Linguistics (Volume 1: Long Papers), pages 861–872, Vancouver, Canada. Association for Computational Linguistics.
>
> Emanuele Bugliarello, Sabrina J. Mielke, Antonios Anastasopoulos, Ryan Cotterell, and Naoaki Okazaki. 2020. It’s easier to translate out of English than into it: Measuring neural translation difficulty by cross-mutual information. In Proceedings of the 58th Annual Meeting of the Association for Computational Linguistics, pages 1640–1649, Online. Association for Computational Linguistics.
>
> Philipp Koehn. 2005. Europarl: A parallel corpus for statistical machine translation. In Proceedings of Machine Translation Summit X: Papers, pages 79–86, Phuket, Thailand.
>
> Vivien Macketanz, Eleftherios Avramidis, Shushen Manakhimova, and Sebastian Möller. 2021. Linguistic evaluation for the 2021 state-of-the-art machine translation systems for German to English and English to German. In Proceedings of the Sixth Conference on Machine Translation, pages 1059–1073, Online. Association for Computational Linguistics
>
>
> __Reason to Reject 2.A) There are various dialects and minority languages. This paper considers African American Language, but it is not clear why the specific language is handled here. The paper may be more appropriate for a focused workshop.__
>
> While there are certainly other minority dialects that may experience similar biases from LLMs, we choose African American Language due to the historical treatment of African Americans in the United States and more specifically, because the dialect has in part reinforced the historical oppression of African Americans through its historical view as a “defective” form of English. With 13.6% of the US population (U.S. Census Bureau, 2020) self-identifying as African American, development of LLMs without consideration of their dialect would exclude a large community of potential users. There is a large body of prior work focusing on AAL in the NLP community and published as main conference papers, many of which we cite in the Introduction and Related Work sections (e.g. Lines 28-30; 539; 540-541; 554-556; 560-561; 566).
>
> U.S. Census Bureau. 2020. 2020 census. U.S. Department of Commerce.
>
>
> __Reason to Reject 2.B) In this paper, "bias" is defined as "gaps in performance between one language and the other language" (L108). It is not clear why such differences are considered as bias, and it can be misleading. This paper considers two simple tasks, but it is unclear if any social impacts can be estimated from the results.__
>
> We note in the Introduction section that performance gaps can reflect representational harms (Line 109-113) and can potentially exacerbate societal inequalities, such as in mental health counseling or medical healthcare (Line 36-57). Additionally, Hovy and Prabhumoye (2021) explicitly mention “unequal performance for different user groups” as a reflection of “demographic bias”, and Shah et al. (2020) defines predictive bias as when “the label distribution of a predictive model reflects a human attribute in a way that diverges from a theoretically defined ‘ideal distribution,’” both of which encompass our analysis of performance gaps. We acknowledge in our limitations section that further work is necessary to evaluate LLMs in realistic use-cases (Lines 608-619), but because little prior work has investigated AAL and LLMs, we chose to perform an intrinsic evaluation as a foundation for future work on the topic.
>
> Dirk Hovy and Shrimai Prabhumoye. 2021. Five sources of bias in natural language processing. Language and Linguistics Compass, 15(8):e12432.
>
> Deven Santosh Shah, H. Andrew Schwartz, and Dirk Hovy. 2020. Predictive biases in natural language processing models: A conceptual framework and overview. In Proceedings of the 58th Annual Meeting of the Association for Computational Linguistics, pages 5248–5264, Online. Association for Computational Linguistics.
>
>
> __Q1) In Figure 2, all the LLMs have comparable performance to Human in BERTScore. Does it mean LLMs are not worse than Human?__
>
> The setting of the counterpart generation task required that the annotators and evaluated models produce semantically equivalent counterparts in the alternate language variety. Additionally, AAL and WME are largely similar language varieties beyond the set of unique features of AAL. So, the lack of differences in semantic similarity measures like BERTScore are expected due to these inherent similarities. While differences between LLMs and Human are small because of these similarities, the other metrics and tasks included do present performance gaps, so we cannot conclude from these results that LLM and Human outputs perform equally well.
>
> __Typos, Grammar, Style And Presentation Improvement 1) In the paper, expressions like "model's interpretation" and "model's understanding" are often used (e.g., L232). They are somewhat confusing. The statistical models just solve optimization problems for given tasks.__
>
> We note that terms such as "natural language understanding" and "interpretation" have been used in the computational linguistics fields for decades in many different contexts (e.g., "semantic interpretation" is a common phrase). We can switch references from "understanding" to "interpretation", but we feel that the solution of an optimization problem for a given input in this case is indeed an interpretation of the input and thus, "interpretation" is an appropriate term.

---

### Official Review · Reviewer_raHm · 2023-08-04

**Soundness:** 4

**Excitement:**

5: Transformative: This paper is likely to change its subfield or computational linguistics broadly. It should be considered for a best paper award. This paper changes the current understanding of some phenomenon, shows a widely held practice to be erroneous in someway, enables a promising direction of research for a (broad or narrow) topic, or creates an exciting new technique.

**Paper Topic And Main Contributions:**

This paper investigates how well current popular LLMs understand African American language (AAL). They evaluate 6 LLMs on two language generation tasks, they also curate a new AAL dataset and perform extensive experiments showcasing the shortcomings of current LLMs on AAL.

**Questions For The Authors:**

- Please report the generation hyperparameters used
- Please report the results for FlanT5 without finetuning
- Why did you use debertalargemnli for BERTScore and not another model?
- How do you reconcile the fact that in the span masking, different spans are being masked so the results are not directly comparable?

**Reasons To Accept:**

- Tackle an important problem of LLM understanding AAL, which is critical for AI inclusion.
- Release a new AAL dataset that should aid research
- Well written with comprehensive methodology that avoids several possible bias
- Important results that show that LLMs struggle with AAL, although finetuning them helps.

**Reasons To Reject:**

- Did not report generation hyperparameters for the LLMs
- Missing important benchmark of FlanT5 without finetuning

**Reproducibility:**

4: Could mostly reproduce the results, but there may be some variation because of sample variance or minor variations in their interpretation of the protocol or method.

**Reviewer Confidence:**

5: Positive that my evaluation is correct. I read the paper very carefully and I am very familiar with related work.

---

> ### Author Rebuttal · Authors · 2023-08-29
>
> Thank you for your review.
>
> __Q1) Please report the generation hyperparameters used__
>
> We will include further details concerning generation hyperparameters use in the appendix and update the corresponding reference in the Methods section. For GPT models, the generation hyperparameters are a temperature of 0.7, and all remaining hyperparameters were left as default (top p of 1, 0 frequency or presence penalty). For all other models (BART, T5, Flan-T5), the generation hyperparameters are 3 beams, a no_repeat_ngram_size of 3, and all other parameters left as default (temperature of 1, no sampling)
>
>
> __Q2) Please report the results for FlanT5 without fine-tuning__
>
> We will update Figures 2, 4, and 5 as well as Tables 6, 8, and 9 in the Appendix to include the results for FlanT5 without fine-tuning. Results for fine-tuned Flan-T5 will be presented as “Flan-T5 (FT)” and without fine-tuning, “Flan-T5.” Across all automatic metrics and in both counterpart generation directions, Flan-T5 without fine-tuning performed slightly worse than with fine-tuning while still presenting performance gaps preferring WME over AAL. Additionally, similar trends were observed concerning toxicity, where Flan-T5 without fine-tuning presented a stronger preference for WME in the non-toxic subset of the data compared to the toxic subset.
>
>
> __Q3) Why did you use debertalargemnli for BERTScore and not another model?__
>
> We use the ‘deberta-large-mnli' checkpoint for BERTScore over other model checkpoints because it has been found to be better correlated with human scores by the original BERTScore authors (Zhang et al., 2019). The associated GitHub repository lists all models’ correlation with human scores. We will update the Metrics section to explicitly state this reasoning.
>
> Tianyi Zhang, Varsha Kishore, Felix Wu, Kilian Q. Weinberger, and Yoav Artzi. 2019. Bertscore: Evaluating text generation with bert.
>
>
> __Q4) How do you reconcile the fact that in the span masking, different spans are being masked so the results are not directly comparable?__
>
> This is an excellent point; we acknowledge that it is difficult to directly compare the masked span prediction results between dialects considering that different spans are masked. Due to the differences in word choice, word order, and other features of AAL, masking identical spans in both texts would be impossible, particularly while attempting to determine if the features of AAL affect model performance. To mitigate this issue, we repeat the masked span prediction experiment 5 times with different randomly masked spans (Lines 330-332), avoid directly comparing metrics at the sample level, and present results for different classes of spans (noun and verb phrases) to allow for more fine-grained and fair comparisons (Lines 228-230). We will add language highlighting the purpose of the repeated experiments in the Methods section.

---

### Meta-Review · Area_Chair_NVKb · 2023-09-18

**Recommendation:** 4

**Metareview:**

The paper does a thorough investigation of the ability of LLMs to understand AAE as compared to SAE. This is a very timely piece of work, as models get more intertwined with lives globally. Discrepancies in quality of service must be investigated, and this paper contributes to this soundly. In addition, they develop a resource for this evaluation. It would be useful to release more details of this such as annotator demographics, compensation etc in a datasheet.
I would also urge the authors to reflect a little however on the definition and usage of the word ‘bias’ in the paper and try to ground it in experienced harms. As noted by a lot of recent literature, the interchangeable or disconnected usage of these words can dilute the impact and utility of a work, and given the thoroughness in this paper, just some additional reflection of the concepts used and framed would amplify its message.

---

### Decision · Program_Chairs · 2023-10-07

**Decision:**

Accept-Main

**Comment:**

The paper does a thorough investigation of the ability of LLMs to understand AAE as compared to SAE. This is a very timely piece of work, as models get more intertwined with lives globally. Discrepancies in quality of service must be investigated, and this paper contributes to this soundly. In addition, they develop a resource for this evaluation. It would be useful to release more details of this such as annotator demographics, compensation etc in a datasheet.
I would also urge the authors to reflect a little however on the definition and usage of the word ‘bias’ in the paper and try to ground it in experienced harms. As noted by a lot of recent literature, the interchangeable or disconnected usage of these words can dilute the impact and utility of a work, and given the thoroughness in this paper, just some additional reflection of the concepts used and framed would amplify its message.